# Global Survey of the Full-Length Cabbage Transcriptome (*Brassica oleracea* Var. *capitata* L.) Reveals Key Alternative Splicing Events Involved in Growth and Disease Response

**DOI:** 10.3390/ijms221910443

**Published:** 2021-09-28

**Authors:** Yong Wang, Jialei Ji, Long Tong, Zhiyuan Fang, Limei Yang, Mu Zhuang, Yangyong Zhang, Honghao Lv

**Affiliations:** 1Institute of Vegetables and Flowers, Chinese Academy of Agricultural Sciences, Key Laboratory of Biology and Genetic Improvement of Horticultural Crops, Ministry of Agriculture, #12 Zhong Guan Cun Nandajie Street, Beijing 100081, China; wangyong03@caas.cn (Y.W.); jijialei@caas.cn (J.J.); nxyxsh.2009@163.com (L.T.); fangzhiyuan@caas.cn (Z.F.); yanglimei@caas.cn (L.Y.); zhuangmu@caas.cn (M.Z.); 2Hengyang Vegetable Research Institute, Huangshawan Street, Shigu District, Hengyang 421000, China

**Keywords:** cabbage, transcriptome, alternative splicing, single-molecule real-time sequencing

## Abstract

Cabbage (*Brassica oleracea* L. var. *capitata* L.) is an important vegetable crop cultivated around the world. Previous studies of cabbage gene transcripts were primarily based on next-generation sequencing (NGS) technology which cannot provide accurate information concerning transcript assembly and structure analysis. To overcome these issues and analyze the whole cabbage transcriptome at the isoform level, PacBio RS II Single-Molecule Real-Time (SMRT) sequencing technology was used for a global survey of the full-length transcriptomes of five cabbage tissue types (root, stem, leaf, flower, and silique). A total of 77,048 isoforms, capturing 18,183 annotated genes, were discovered from the sequencing data generated through SMRT. The patterns of both alternative splicing (AS) and alternative polyadenylation (APA) were comprehensively analyzed. In total, we detected 13,468 genes which had isoforms containing APA sites and 8978 genes which underwent AS events. Moreover, 5272 long non-coding RNAs (lncRNAs) were discovered, and most exhibited tissue-specific expression. In total, 3147 transcription factors (TFs) were detected and 10 significant gene co-expression network modules were identified. In addition, we found that Fusarium wilt, black rot and clubroot infection significantly influenced AS in resistant cabbage. In summary, this study provides abundant cabbage isoform transcriptome data, which promotes reannotation of the cabbage genome, deepens our understanding of their post-transcriptional regulation mechanisms, and can be used for future functional genomic research.

## 1. Introduction

Cabbage (*Brassica oleracea* L. var. *capitata* L.) originates from the Mediterranean region. It is an important biennial vegetable belonging to the genus Brassica and mustard family, Brassicaceae (Cruciferae). Many nutrients such as vitamin C, vitamin K, sulforaphane, and indole-3-carbinol are found in cabbage [1]. According to the Food and Agriculture Organization (FAO) statistics, total global cabbage and other brassicas 2018 harvest area was 2.41 million hm^2^, of which about 0.98 million hm^2^ were grown in China.

The whole genome sequencing of cabbage provides an important reference for gene mapping, cloning, and molecular marker development. The reported cabbage genome size is about 530 Mb and approximately 45,000 protein-coding genes were annotated [2,3]. With the rapid developing of high-throughput sequencing techniques, transcriptome sequencing is widely adopted by researchers to study gene expression regulation. Previous studies on gene transcripts were primarily based on the sequence data generated by next-generation sequencing (NGS) technology. Although NGS technique greatly improves sequencing data and enables the discovery of novel transcripts, NGS reads are short and have had unsatisfactory results in transcript structure analysis. These issues are especially highlighted in the detection of new transcripts, alternative splicing (AS), and alternative polyadenylation (APA).

AS and APA, widely known as two significant patterns of post-transcriptional regulation, can greatly improve the diversity level of the transcipts and proteins [4,5,6]. AS, which has been found to be ubiquitous in eukaryotes, refers to the phenomenon that a single pre-mRNA generates multiple transcript isoforms based on different splicing sites [6,7]. Genome-wide studies have displayed that more than 60% of intron-containing genes in plants possess different splicing sites [6]. Moreover, some research has suggested that environmental and biotic stressors can induce or greatly impact the AS events in plants [8,9,10]. For instance, when *Arabidopsis thaliana* is infected by *Pseudomonas syringae*, more than 90% of the expressed genes (23,385 out of 25,619) undergo AS [11]. Similarly, APA can also result in multiple transcript isoforms with various 3′ ends [5,12]. Genome-wide analyses suggest that approximately 48% of expressed genes from rice and 70% of genes from *A. thaliana* possess multiple polyadenylation sites [5,13]. AS and APA are two significant post-transcriptional patterns of regulation, however, they have not been revealed in cabbage yet.

With the rapid development and application of third-generation sequencing platforms, now the average length of sequenced reads can reach 10 kb and with a maximum length that can exceed 40 kb. These average read lengths can cover most typical genes. Therefore, they can make up for the limitations of NGS read lengths. The third-generation sequencing technology used in this paper was PacBio RS II Single-Molecule Real-Time (SMRT) sequencing. Full-length transcriptome sequencing technology was adopted using SMRT to facilitate the identification of AS, APA, and new transcripts.

Prior to this study, each cabbage gene was annotated with only one transcript, while information of post-transcriptional regulations such as AS, APA, and other isoforms are largely unknown. In the current study, the SMRT sequencing technique was adopted to analyze the comprehensive cabbage transcriptome from five different cabbage tissues. A total of 77,048 isoforms, 5272 long non-coding RNAs (lncRNAs), 13,468 APA genes, and 8978 AS genes at genome wide level were identified. The divergence of AS events in various cabbage tissues and the AS response to Fusarium wilt infection was also analyzed. These results provide a valuable resource for improving the current cabbage genome annotation and exploring the post-transcriptional mechanisms in cabbage genes.

## 2. Results

### 2.1. Full-Length Transcriptome Sequencing Analysis

To comprehensively characterize the cabbage transcriptome profile, five tissues, including root, stem, leaf, flower, and silique, were used for full-length transcript sequencing using the PacBio Iso-Seq platform. A total of 51.3 Gb subreads were obtained and used for transcriptome analysis after filtering through SMRT Link. As shown in Appendix A, we obtained 1,917,514 circular consensus sequencing (CCS) reads using the Iso-Seq protocol. A total of 1,381,218 CCS reads were full-length non-chimeric reads (FLNCs). The average length of FLNCs was 1971 bp. Then, soform-level clustering (ICE) was used to improve consensus sequence accuracy and remove redundant FLNCs with identical transcripts sequence. Finally, a total of 669,351 consensus reads were generated. The most consensus reads were obtained from root tissue (177,861 reads), followed by stem (164,425 reads), silique (119,533 reads), flower (113,545 reads), and leaf (93,987 reads). In addition, we also used accurate short reads produced from NGS in this study to correct the consensus reads from PacBio using the LoRDEC algorithm (Appendix A).

The corrected consensus sequences were mapped to the cabbage reference genome (D134) using GMAP. More than 97% of reads were mapped to the reference genome (Table 1). According to the mapping results, the reads were divided into five categories which were as follows: unmapped reads, multiple mapped reads, reads mapped to ‘+’, and reads mapped to ‘−’. The unmapped reads account for approximately 2% of consensus sequences, showing no significant mapping to the reference genome. The multiple mapped reads showed multiple alignments and account for about 8% of the consensus sequences. Reads mapped to the ‘+’ strand were mapped to the positive strand of the genome and account for about 70% of consensus sequences. Reads mapped to ‘−’ were mapped to the opposite strand of the genome and account for about 20% of the consensus sequences.

A total of 77,048 isoforms were found to cover 18,183 genic loci. As shown in Figure 1A, 9798 isoforms were shared across all tissues and 29,451 (38.22%) isoforms were tissue specific. A great number of novel isoforms in known genes were identified across the five tissues, suggesting AS, APA, or other post-transcriptional mechanisms may play important roles in cabbage development (Figure 1B). Moreover, we detected that approximately 7% of the isoforms were generated from novel genes. The GO functional annotations and KEGG analysis of tissue-specific genes are shown in Figure 1C,D. Silique-specific genes were mainly involved in “starch and sucrose metabolism”, “pentose and glucuronate interconversions”, and “amino sugar and nucleotide sugar metabolism”.

### 2.2. Analysis of Alternative Splicing (AS)

To analyze the AS events in cabbage, we used SpliceGrapher [14,15] to characterize splicing variants among different cabbage tissues based on the Iso-Seq data. AS events mostly include the following types: skipped exons (SE), mutually exclusive exons (MX), alternative 5′ splice sites (A5), alternative 3′ splice sites (A3), intron retention (IR), alternative first exon (AF), and alternative last exon (AL). The results indicated that a total of 8978 AS genes were identified from 18,183 annotated genes in the five cabbage tissues. Among the 8978 AS genes, only 436 AS genes were found to be shared across all tissues. However, approximately 50% of AS genes were tissue specific (Figure 2A). The number of AS genes found in roots was the largest, followed by stems, leaves, flowers, and siliques (Figure 2B). Among the AS types, intron retention (IR) occurred most frequently, accounting for about 50% of the AS events (Figure 2C). The distribution of all AS types across the different tissues is shown in Figure 2C. To verify the AS events, primers spanning the AS regions of several genes were designed and reverse transcription PCR (RT-PCR) was performed. As shown in Figure 2D, the gel band fragment sizes were in accordance with the spliced isoforms detected from the SMRT sequencing data.

### 2.3. Identification of lncRNAs

Long noncoding RNAs (lncRNAs) are a class of transcribed RNAs that exceed 200 nucleotides in length but do not encode proteins. Several studies have shown that lncRNAs can modulate key molecular and biological processes, such as regulation of transcription and gene silencing. In this study, we applied three tools, CNCI [16], CPC [17], and Pfam [18], to identify lncRNAs from single-molecule long-read (SMLR) isoforms. Based on the results, 5272 putative lncRNAs were obtained. As shown in Figure 2A, only 14 lncRNAs were shared across all tissues. However, 4301 lncRNAs were found to be tissue specific. The length of these lncRNAs ranged from 200 to 5819 bp. According to analyses, lncRNAs with a length of 200–500 bp account for 76% of lncRNAs. The average length of lncRNAs was 478 bp. Mapping lncRNAs to the genome showed that 60.8% of lncRNAs were generated from intergenic regions, 13.6% of lncRNAs were generated from sense strands, 13.6% of lncRNAs were generated from antisense strands, and 12.0% of lncRNAs were generated from intronic regions.

### 2.4. Analysis of Alternative Polyadenylation (APA)

In addition to AS, APA can also improve the diversity of transcriptome and proteome by generating transcript isoforms with different 3′ ends. Studying the 3′ ends of transcripts using SMRT sequencing allowed us to identify polyadenylation sites in cabbage. In this study, two computational methods, PRAPI [19] and TAPIS [20], were applied to identify APA events using cabbage Iso-Seq data. In total, 13,468 genes with poly(A) sites were identified from five different tissues. As shown in Figure 3B, only 199 APA genes were shared across all tissues. However, 7384 APA genes were found to be tissue specific. A number of genes were found to have at least six poly(A) sites in roots (92 genes), stems (128 genes), leaves (27 genes), flowers (50 genes), and siliques (11 genes) (Figure 3C).

### 2.5. Analysis of Transcription Factors

Transcription factors (TFs) are proteins that bind to DNA-regulatory sequences (enhancers and silencers) in order to regulate gene expression. In this study, we applied iTAK software to identify 3147 TFs using the Iso-Seq data. A signed weighted gene co-expression network analysis of all TFs was conducted using weighted correlation network analysis (WGCNA). A total of 10 significant co-expression modules were found to be related to the expressed TFs (Figure 4A). The yellow, black, green, brown, and magenta modules were identified as the top five significant modules in the co-expression network associated with the development of five tissues. Among these five modules, the magenta module contained only 58 TFs, which is the module with fewest TFs. Most of the TFs contained in the magenta module are highly expressed in siliques (Figure 4B). Moreover, most of the TFs contained in the black, brown, yellow, and green modules were highly expressed in stem, flower, root, and leaf tissues, respectively (Figure 4B).

The top 10 genes in the black, brown, green, magenta, and yellow modules all contained Zinc-finger, MYB, WRKY, NAC, Myc-type, and Leucine type TFs (Figure 4C). In addition, most TFs related to signal transduction were highly expressed in stem tissue. Most GNAT and BTB/POZ TFs showed expression peaks in flower tissue. Most TFs related to auxin showed expression peaks in root, leaf, flower, and silique tissues. Most AP2/ERF TFs were mainly expressed in root, silique, and stem tissues.

### 2.6. Identification of Differential AS Variants in Response to Fusarium Wilt, Black Rot and Clubroot

Previous research has suggested that AS plays an important role in strengthening plant disease resistance by regulating gene (especially the resistance (R) genes) expression and increasing the isoform diversity in the transcriptome and proteome [8]. To investigate cabbage splicing variants in response to Fusarium wilt (FW), RNA-Seq data [21] of cabbage plants that were inoculated with *Fusarium oxysporum* f. sp. *conglutinans* (Foc) were used to analyze AS events. From our results, the amount of differential AS events (DAS) in the FW-resistant cabbage line increased significantly after inoculation with Foc, especially at nine days post-inoculation (dpi) (Figure 5A,B). However, there was no significant difference of AS in FW-susceptible cabbage plants across different times after Foc inoculation. There were 666 DAS genes that were shared in all groups of the FW-resistant cabbage, and 380 DAS genes that were specific at nine dpi (Figure 5A). However, there were only 80 and 46 DAS genes that were specific at three dpi and six dpi, respectively (Figure 5A). Moreover, three R genes (NBS-LRR type genes) (*Boc04g03962*, *Boc03g06003*, and *Boc03g00010*) were identified as the nine-dpi-specific DAS genes (Figure 5C), indicating that these R genes may be involved in the regulation of resistance to FW.

Moreover, splicing variants of cabbage in response to black rot (BR) and clubroot (CR) were also analyzed using RNA-Seq data [22,23]. We found 1826 and 2424 DAS genes in BR-resistant and CR-resistant lines, respectively. Interestingly, 473 DAS genes were shared in FW-resistant, BR-resistant and CR-resistant lines (Figure 5D), suggesting these genes may be related to plant disease resistance.

## 3. Discussion

In the past decade, NGS technology has been widely used in eukaryotic transcriptomic analysis. However, the short-reads generated by NGS have their own limitations for accurate transcript assembly. Emerging long-read Iso-Seq technology can be used to generate full length isoforms, which facilitate the comprehensive analysis of entire transcriptomes at the isoform level. In this study, PacBio Iso-Seq technology was used to produce an overall full-length transcriptome database, which contains full length isoforms of five cabbage tissues.

In the current cabbage genome annotation, only a single transcript is annotated for each gene. In this study, 28,510, 24,369, 30,452, 30,986, and 20,122 unannotated isoforms were identified from flower, silique, root, stem, and leaf tissues, respectively. Approximately 90% of these new isoforms were generated from known genes. A total of 5272 high-confidence lncRNAs were identified and most of the lncRNAs were found to be tissue specific. In addition, a total of 3147 TFs were detected using iTAK software, and 10 significant gene co-expression modules between all TFs were identified by WGCNA. These results will significantly supplement the current cabbage genome annotation.

The Iso-Seq data were also used for analysis of cabbage APA at the whole-genome level. From our results, we detected 13,468 genes that can generate different isoforms through APA. The differences of APA among different tissues were also characterized. These results deepen our understanding of post-transcriptional regulatory mechanisms in cabbage. For the AS analysis, we detected a total of 8978 AS genes and most of the AS events were of the intron retention (IR) type. Only 5% of AS events were found to be shared across the five tissues, whereas about 50% were tissue specific, indicating that AS may play important roles in the regulation of certain tissue-specific functions. These results suggest that full-length reads generated from Iso-Seq can help identify AS events or complex isoforms without the aid of sequencing assembly. In conclusion, SMAT sequencing is an important technology for isoform identification and transcriptional investigation.

AS leads to polymorphisms in the structure and function of transcripts and proteins. It is an important transcriptional regulatory mechanism that can cope with the infection of a variety of pathogens in plants [11,24,25,26]. Many plant R genes undergo AS, and some R genes require alternately spliced transcripts to produce R proteins that specifically recognize pathogen invasion. In the fine regulation process of R protein activation, the truncated R protein subtype produced by AS may participate in plant disease resistance by inhibiting the negative regulation of immune initiation or directly participating in signal transduction triggered by effector factors [27].

After the *A. thaliana* was inoculated with *Pseudomonas syringae*, the exons in the plant increased by more than 44%, which provided evidence for the pathogen to induce AS in plants [27]. Further studies have shown that overexpression of the RNA binding protein GRP7 leads to increased resistance to *Pseudomonas syringae* and changes the shearing of PR (Pathogenesis-Related gene) transcripts related to salicylic acid and jasmonic acid-dependent defense, and regulates different PR transcripts expression [28,29].

Additionally, cabbage splicing variants in response to Fusarium wilt, black rot and clubroot were investigated. A total of 1250, 1826 and 2424 genes were found to be induced to undergo AS events after inoculation with Foc, *Xanthomonas campestris* pv. *campestris* and *Plasmodiophora brassicae*, respectively. Specifically, 473 DAS genes were shared in FW-resistant, BR-resistant and CR-resistant lines, indicating that AS may help cabbage adapt to adverse environments. Generally, the resources generated from this study will promote future studies in cabbage post-transcriptional regulation mechanisms.

## 4. Materials and Methods

### 4.1. Plant Material and RNA Sample Preparation

The cabbage double-haploid line D134, which shows excellent horticultural characteristics, was used in this study. The plant materials were grown at the Institute of Vegetables and Flowers, Chinese Academy of Agricultural Sciences, Beijing, China. The leaf, stem, and root tissue were collected at the four-leaf stage. Flowers and siliques with the length of 3–4 cm were gathered at the later flower stage. All samples were quickly frozen in liquid nitrogen and stored at −80 °C.

### 4.2. RNA Extraction and Library Preparation

The total RNA of samples from different tissues was extracted by the Trizol method. The RNA quality was evaluated by agarose gel electrophoresis, Nanodrop, Qubit, and Agilent 2100. The RNA sample with a RIN (RNA integrity number) value of more than 8 is used to construct a Pacbio sequencing library. The mRNA was enriched using magnetic beads with Oligo (dT). Then the mRNA was reverse transcribed into cDNA using the SMARTer PCR cDNA Synthesis Kit. The BluePippin was used for fragment screening. The full-length cDNA was subjected to damage repair, end repair, SMRT linker ligation, and exonuclease digestion to obtain a library.

### 4.3. Full-Length Transcriptome Sequencing and Data Analysis

The library sequencing was performed using the Pacbio Sequel platform. The raw data is processed by the software package SMRTlink to obtain the subreads sequence. The circular consensus sequence (CCS) is generated through correction of subreads. According to whether the sequence contains 5′- or 3′-end primer and polyA tail, the sequences are divided into full-length sequences and non-full-length sequences. The full-length sequences were clustered using isoform-level clustering (ICE) to obtain the cluster consensus sequences. The non-full-length sequences were used to correct the consensus sequences to generate high-quality sequences for subsequent analysis. Additional nucleotide errors in consensus reads were corrected using the Illumina RNA-seq data with the software proovread. The GMAP was used to align consensus reads to the cabbage reference genome.

### 4.4. Illumina RNA-Seq and Data Analysis

The quality of RNA from the three biological replicates of each tissue was evaluated by agarose gel electrophoresis, Nanodrop, Qubit, and Agilent 2100. The RNA samples with OD260/280 ≥ 1.8, OD260/230 ≥ 1.8 and 28S/18S ≥ 1.0 were selected for constructing Illumina RNA-seq libraries. The cDNAs of all samples were sequenced using the HiSeq 4000 PE150 platform. The NGSQCToolkit software [30] was used to eliminate the pair-end reads that contain more than 5% unknown nucleotides or 20% of bases with QS (quality score) < 20. TopHat [31] was used to align the high-quality reads to the cabbage reference genome. Saturation analysis was performed to reduce the transcription noise. The assembled transcripts generated by Trinity [32] were mapped to the cabbage reference genome using GMAP [33].

### 4.5. Coding Potential Analysis

To distinguish protein-coding and non-coding sequences, CNCI (Coding-Non-Coding-Index) [16] with default parameters was used to analyze adjoining nucleotide triplets of transcripts. CPC (Coding Potential Calculator) [17] was used to assess the extent and quality of ORFs in a transcript and search the sequences with the NCBI eukaryotes’ protein database to clarify the coding and non-coding transcripts. Moreover, each transcript was translated in all three possible frames and the Pfam Scan [18] was used to identify the occurrence of any known protein family domains documented in the Pfam database.

### 4.6. Characterization of AS Events

The SpliceGrapher [14,15] and rMATS [12] were used to analyze differential splicing events. The examined events included skipped skipped exons (SE), mutually exclusive exons (MX), alternative 5′ splice sites (A5), alternative 3′ splice sites (A3), intron retention (IR), alternative first exon (AF), and alternative last exon (AL). Statistical analysis was performed using the in-house Perl script. To investigate cabbage splicing variants in response to Fusarium wilt, black rot and clubroot, RNA-Seq data [21,22,23] of cabbage plants that were inoculated with *Fusarium oxysporum* f. sp. *conglutinans*, *Xanthomonas campestris* pv. *campestris* and *Plasmodiophora brassicae* Woronin were used to analyze AS events.

### 4.7. Identification of APA and Unannotated Genes

APA sites and unannotated genes were identified using PRAPI [19] and TAPIS [20] and only coincident results identified from both of them were used for further analyses. The MEME-ChIP analysis was performed based on the sequence of 50 nucleotides upstream of the poly(A) sites to identify the poly (A) sequence signals. We searched for homologs of unannotated genes in UniProt and Ensembl Plants databases through BLASTX and TBLASTX analysis with the E value threshold of 1 × 10^−6^.

### 4.8. Validation by RT-PCR

Total RNA from the same sample as described above for SMAT sequencing was extracted by the Trizol method. The RNA was reverse transcribed into cDNA using the PrimeScript^TM^ RT Reagent Kit (Takara). The primers spanning the splicing events were designed through PRIMER PREMIER software (v.5.0). The Q5 high-fidelity DNA polymerase was used for RT-PCR. The PCR product was analyzed following electrophoresis in a 2% agarose gel.

## Figures and Tables

**Figure 1 ijms-22-10443-f001:**
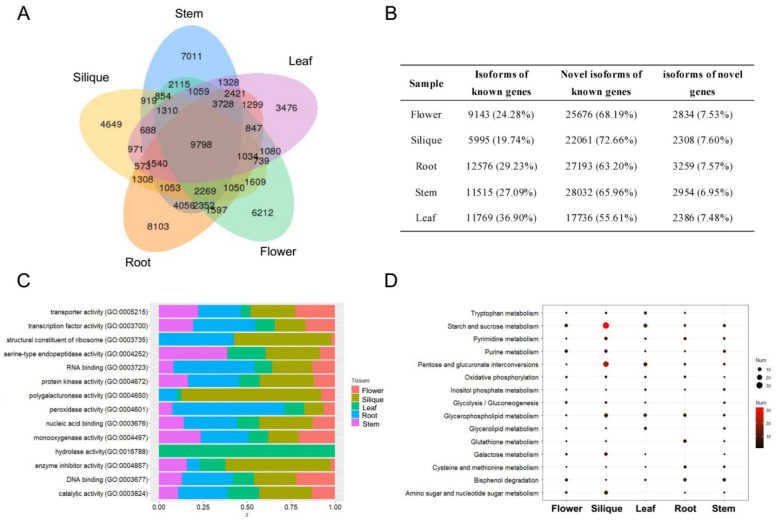
Statistics and annotations of the isoforms obtained by SMRT sequencing. (**A**) Venn diagram illustrating unique and common isoforms in different tissues. (**B**) Statistics of the isoforms in different tissues. (**C**) GO annotations of the tissue-specific genes. (**D**) KEGG annotations of the tissue-specific genes.

**Figure 2 ijms-22-10443-f002:**
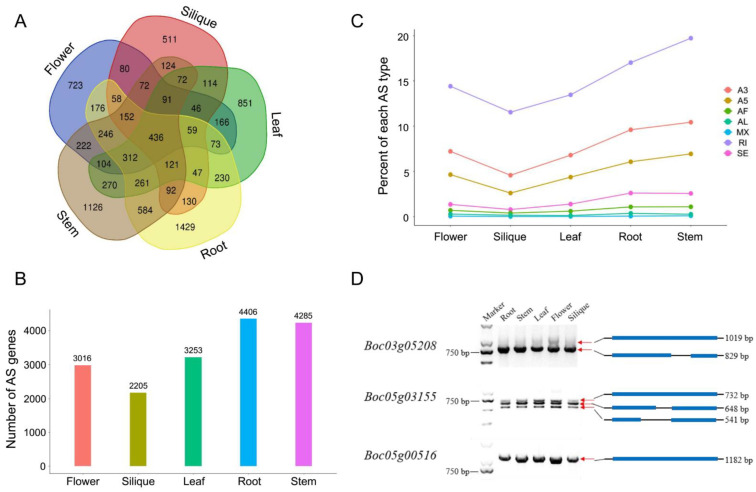
Characterization of genes with alternative splicing (AS). (**A**) Venn diagram illustrating unique and common AS genes in different tissues. (**B**) Number of AS genes in different tissues. (**C**) Percent of each AS type in different tissues. (**D**) Verification of AS events by RT-PCR.

**Figure 3 ijms-22-10443-f003:**
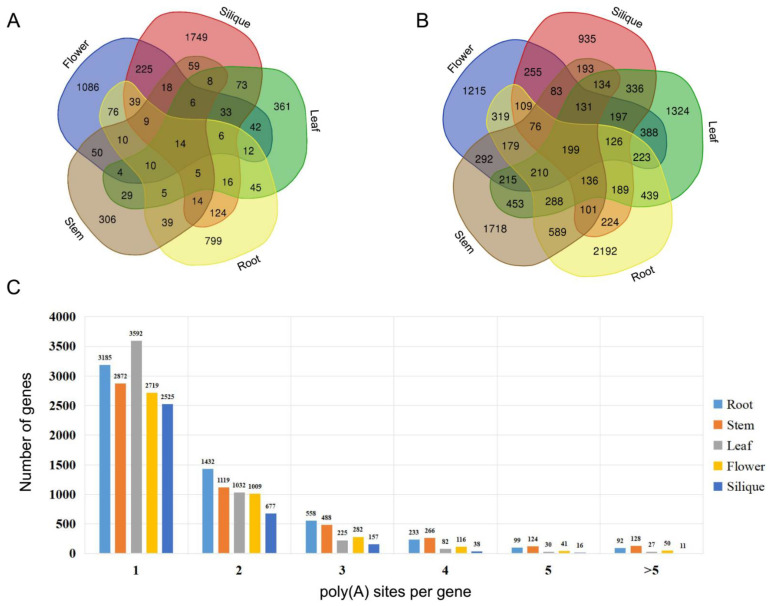
Statistics for long noncoding RNAs (lncRNAs) and genes with alternative polyadenylation (APA). (**A**) Venn diagram illustrating unique and common lncRNAs in different tissues. (**B**) Venn diagram illustrating unique and common APA genes in different tissues. (**C**) Number of the genes with different poly(A) sites in five tissues.

**Figure 4 ijms-22-10443-f004:**
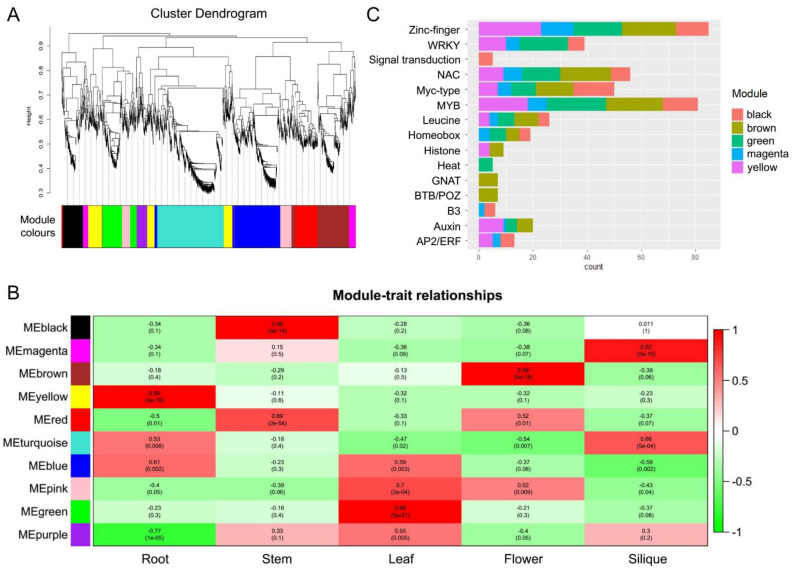
Weighted correlation network analysis of transcription factors (TFs). (**A**) Cluster dendrogram of the expressed TFs. (**B**) Heat map of module–trait relationships. (**C**) TF types in different co-expression modules.

**Figure 5 ijms-22-10443-f005:**
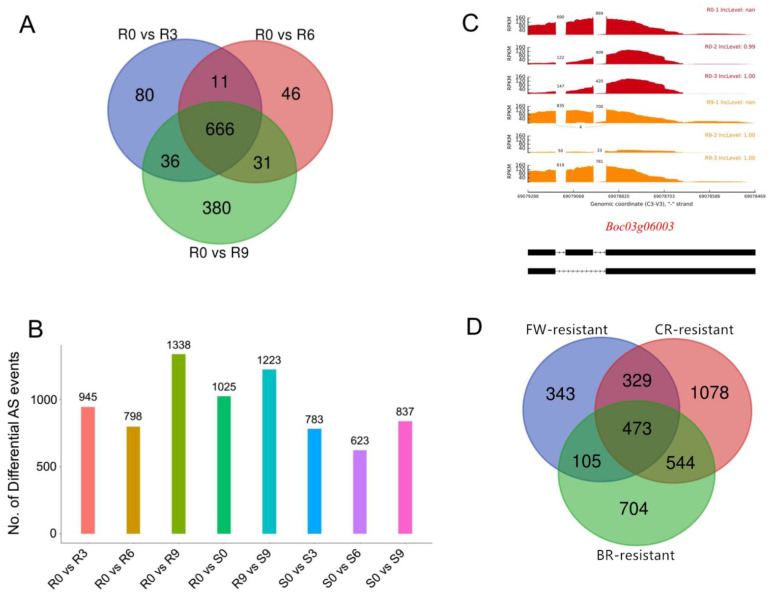
Analysis of the cabbage splicing variants in response to Fusarium wilt (FW). (**A**) Venn diagram illustrating unique and common AS genes at different times after *Fusarium oxysporum* f. sp. conglutinans (Foc) inoculation. (**B**) Number of differential AS events at different times after Foc inoculation. (**C**) The splicing variant of *Boc03g06003* in response to FW. R0, R3, R6, and R9 represent resistant plants at 0, 3, 6, and 9 dpi, respectively. S0, S3, S6, and S9 represent susceptible plants at 0, 3, 6, and 9 dpi, respectively. (**D**) Venn diagram illustrating unique and common AS genes in response to black rot (BR), clubroot (CR) and Fusarium wilt (FW).

**Table 1 ijms-22-10443-t001:** The mapping results with GMAP.

Sample	Unmapped Reads	Multiple Mapped Reads	Reads Mapped to ‘+’	Reads Mapped to ‘−’
Flower	2078 (1.83%)	10,772 (9.49%)	76,449 (67.33%)	24,246 (21.35%)
Silique	3140 (2.63%)	18,050 (15.10%)	71,957 (60.20%)	26,386 (22.07%)
Root	3414 (1.92%)	7828 (4.40%)	131,403 (73.88%)	35,216 (19.80%)
Stem	3135 (1.91%)	10,600 (6.45%)	117,706 (71.59%)	32,984 (20.06%)
Leaf	2350 (2.50%)	6561 (6.98%)	70,326 (74.83%)	14,750 (15.69%)

## Data Availability

The Iso-Seq raw data were submitted to China National GeneBank DataBase with the accession number CNP0001459.

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
