# Peer review of "Global Survey of the Full-Length Cabbage Transcriptome (Brassica oleracea Var. capitata L.) Reveals Key Alternative Splicing Events Involved in Growth and Disease Response"

_ijms, 2021, doi:10.3390/ijms221910443_

Round 1
Reviewer 1 Report
The experimental designing and execution are good, for which I appreciate the authors. The outputs of this study going to be a great resource for researchers working in Brassica crops. However, it seems the raw sequencing data from this study was not made available to public databases like NCBI SRA or others. Hence I request the authors to submit the same. The overall work is found to be suitable for publication with minor revision as mentioned below.
Figure 1,4B and 5C quality are poor and it should be replaced with a good one.
The organism names like A. thaliana and Pseudomonas syringae should be italicized.
Author Response
Comment: The experimental designing and execution are good, for which I appreciate the authors. The outputs of this study going to be a great resource for researchers working in Brassica crops. However, it seems the raw sequencing data from this study was not made available to public databases like NCBI SRA or others. Hence I request the authors to submit the same. The overall work is found to be suitable for publication with minor revision as mentioned below.
Figure 1,4B and 5C quality are poor and it should be replaced with a good one.
The organism names like A. thaliana and Pseudomonas syringae should be italicized.
Response: We are very sorry for our unclear report in these points. The raw sequencing data were submitted to China National GeneBank DataBase with the accession number CNP0001459. However, the data disclosure date is May 1, 2022. Figure 1,4B and 5C was replaced with a good one. The organism names like A. thaliana and Pseudomonas syringae have been italicized.
Reviewer 2 Report
This research suggests that the next-generation sequencing (NGS) technology is satisfied in transcript structure analysis especially at new transcripts, alternative splicing (AS), and alternative polyadenylation (APA). Therefore, they use PacBio RS II Single-Molecule Real-Time (SMRT) sequencing, the third-generation sequencing technology, which can read over 40 kb. These average read lengths can cover most typical genes. In addition, they found that Fusarium wilt, black rot and clubroot infection significantly influenced AS in resistant cabbage. So this study providing abundant cabbage isoform transcriptome data, which promotes reannotation of the cabbage genome, deepens our understanding of their post-transcriptional regulation mechanisms and can be used for future functional genomic research.
Points in favor:
This paper has good significance in point of overcoming the limitations of existing technology, also it is meaningful that the new generation sequencing technology(SMRT) can be applied to any genes not only cabbage but also others. It can contribute to various experiments and they explain well about alternative splicing (AS), alternative polyadenylation (APA) which can be detected by the next-generation sequencing (NGS). So, it makes readers understand easily what they want to figure out.
Points against:
These analysis data are almost quantitive data. So they can show correlation or amount of genes in each tissue type. But, it can’t contain its own analysis of each gene.
Explanation about Boc04g03962, Boc03g06003, and Boc03g00010 is not described until paragraph 2.6. It could be great to annotate what is those genes in the introduction.
The method of disease analysis is not enough. It could be better to notice details.
Author Response
Comment 1: These analysis data are almost quantitive data. So they can show correlation or amount of genes in each tissue type. But, it can’t contain its own analysis of each gene.
Response: We are very sorry for our unclear report in this. The dataset containing the annotations for each gene has been uploaded to China National GeneBank DataBase with the accession number CNP0001459. In the future, we will conduct more in-depth research on some genes of interest.
Comment 2: Explanation about Boc04g03962, Boc03g06003, and Boc03g00010 is not described until paragraph 2.6. It could be great to annotate what is those genes in the introduction.
Response: It is our negligence and we are sorry about this. We added that these genes are NBS-LRR type genes in paragraph 2.6. At present, the biological functions of these genes are not yet clear. In the future, we will conduct a comprehensive study of the functions of these genes..
Comment 3: The method of disease analysis is not enough. It could be better to notice details.
Response: We are very sorry for our unclear report in this. We have supplemented the method of disease analysis in paragraph 4.6.
We tried our best to improve the manuscript and all changes in the manuscript are marked in red.
At last, I want to thank you sincerely for your suggestions and I feel so sorry that so much of your precious time was wasted on our paper revision.